# Seasonal dependence of the Earth's radiation belt: new insights

Rajkumar Hajra[1]

[1]Indian Institute of Technology Indore, Simrol, Indore 453552, India

**Correspondence:** Rajkumar Hajra (rajkumarhajra@yahoo.co.in)

**Abstract.** Long-term variations of the relativistic ($\sim$MeV) electrons in the Earth's radiation belt are explored to study seasonal features of the electrons. An L-shell dependence of the seasonal variations of the electrons is reported for the first time. A clear $\sim$6-month periodicity, representing one/two peaks per year, is identified for 1.5-6.0 MeV electron fluxes in the L-shells between $\sim$3.0 and $\sim$5.0. The relativistic electron flux variation is strongest during solar cycle descending to minimum phases, with weaker/no variations during solar maximum. If two peaks per a year occur, they are largely asymmetric in amplitude. The peaks generally do not have an equinoctial dependence. Sometimes the peaks are shifted to solstices and sometimes one annual peak is only observed. No such seasonal features are prominent for L < 3.0 and L > 5.0. The results imply varying solar/interplanetary drivers of the radiation belt electrons at different L-shells. This has a potential impact on the modeling of space environment. Plausible solar drivers are discussed.

## 1 Introduction

Earth-orbiting satellites traversing through the radiation belts (Van Allen et al., 1958) are vulnerable to relativistic ($\sim$MeV) electrons that can cause internal charging leading to satellite component damage or even satellite loss in extreme cases (e.g., Wrenn, 1995; Iucci et al., 2005; Horne et al., 2013; Baker et al., 2018, and references therein). The MeV electrons in the outer radiation belt (L > 2.5) are known to be accelerated from the $\sim$10-100 keV (energetic) electrons which are injected into the nightside magnetosphere by substorms (e.g., DeForest and McIlwain, 1971; Horne and Thorne, 1998) and convection events in high-intensity long-duration continuous AE activities (HILDCAAs; Tsurutani et al., 2004). The temperature anisotropy of the electrons leads to plasma instability generating whistler-mode chorus waves (Kennel and Petschek, 1966; Tsurutani and Smith, 1974). Resonant interaction of the $\sim$100 keV electrons with chorus waves leads to MeV electron acceleration (Inan et al., 1978; Horne and Thorne, 2003; Summers et al., 2007; Tsurutani et al., 2013; Xiao et al., 2014; Foster et al., 2017; Matsui et al., 2017; Omura et al., 2019; Zhang et al., 2020).

From the above scenario, it implies that the injections of seed ($\sim$10-100 keV) electrons through substorms/HILDCAAs along with electron loss processes control the variability of magnetospheric MeV electrons in the outer radiation belt. In other words, the solar wind-magnetosphere coupling processes that cause substorms and HILDCAAs, play an important role in MeV

electron variability. The electrons are reported to vary in the time scales of a fraction of a second (e.g., microbursts; Tsurutani et al., 2013) to several years. While short-scale variations are attributed to wave-particle interactions and associated solar wind and interplanetary variations, long-term variations are associated with solar activity cycle (e.g., Baker et al., 1986; Tsurutani et al., 2006, 2016; Miyoshi and Kataoka, 2011; Hajra et al., 2013, 2014a, b, 2015a, b, 2020; Li et al., 2015; Hajra and Tsurutani, 2018). Several studies of MeV electrons (e.g., Baker et al., 1999; Li et al., 2001; Kanekal et al., 2010) reported strong semi-

annual modulations of the electrons, and discussed this in the context of the Earth's position in the heliosphere (Cortie, 1912), relative angle of solar wind incidence with respect to Earth's rotation axis (Boller and Stolov, 1970), and geometrical controls of interplanetary magnetic fields (Russell and McPherron, 1973). The aim of this present work is a critical exploration of the seasonal features of the MeV electrons, and to identify their solar activity and L-shell dependencies, if any.

## 2 Data analysis and Results

Figure 1 (top panel) shows the variation of the monthly mean differential fluxes of the electrons in the energy range between 1.5 and 6.0 MeV in different L-shells from 0.5 to 8.5 from July 1992 through June 2004. The L parameter is the radial distance in Earth radii at the equator for a dipole approximation of the Earth's magnetic field (McIlwain, 1961). A L* parameter (Roederer, 1970) could have been used, but because most of the primary results pertain to L < 5.0, it is felt that the L parameter is reasonable to use for this effort. The electron observations are made by the Solar, Anomalous, and Magnetospheric Particle

Explorer (SAMPEX; Baker et al., 1993) that monitored the radiation belts from a low-altitude ($\sim$520-670 km), highly ($82°$) inclined orbit. Figure 1 shows a classical picture of the Van Allen radiation belts: an inner belt with lower fluxes of 1.5-6.0 MeV electrons at L < 2.0, separated by a slot region devoid of any electrons up to L $\sim$2.5, followed by an outer belt extending up to L $\sim$7.0. Peak fluxes occur around L $\sim$3.0-4.5, shown by superposed black lines.

The MeV electron flux variations are compared with monthly mean solar wind speed Vsw (Figure 1 (second panel), red

curve, legend on the right), percentage occurrences of days with daily peak Vsw $\geq$ 500 $\mathrm{km\,s^{-1}}$ (marked as D500, Figure 1 (second panel), histograms, legend on the left), monthly mean solar wind electric field VBs (Figure 1 (third panel)) where V represents Vsw, and Bs is the southward component of interplanetary magnetic field (IMF) or is zero in absence of southward component. VBs has been shown to be the main driver of geomagnetic activity (e.g., Burton et al., 1975; Tsurutani et al., 1992, 1995; Finch et al., 2008). The bottom panel (Figure 1) shows the monthly mean F10.7 solar flux that depicts the $\sim$11-year

solar activity cycle. The period under study extends from the descending phase of solar cycle 22 to the descending phase of solar cycle 23. The solar wind and IMFs are obtained from the OMNI website (https://omniweb.gsfc.nasa.gov/). The OMNI database is formed by time-shifting the observations made by the NASA's ACE, Wind and IMP 8 spacecraft to the Earth's bow shock nose. The IMFs in geocentric solar magnetospheric (GSM) coordinates are used in this work.

An overall association of solar wind high-speed (Vsw $\geq$ 500 $\mathrm{km\,s^{-1}}$) streams (HSSs) and MeV electron fluxes can be

observed from the figure. Both Vsw and D500 exhibit two prominent peaks, one around 1994-1995 and another around 2003-2004, which are in the descending phases of solar activity cycles 22 and 23, respectively. These intervals are characterized by flux enhancements and broadening of the 1.5-6.0 MeV electron belt in L-shell space. On the other hand, a clear narrowing of

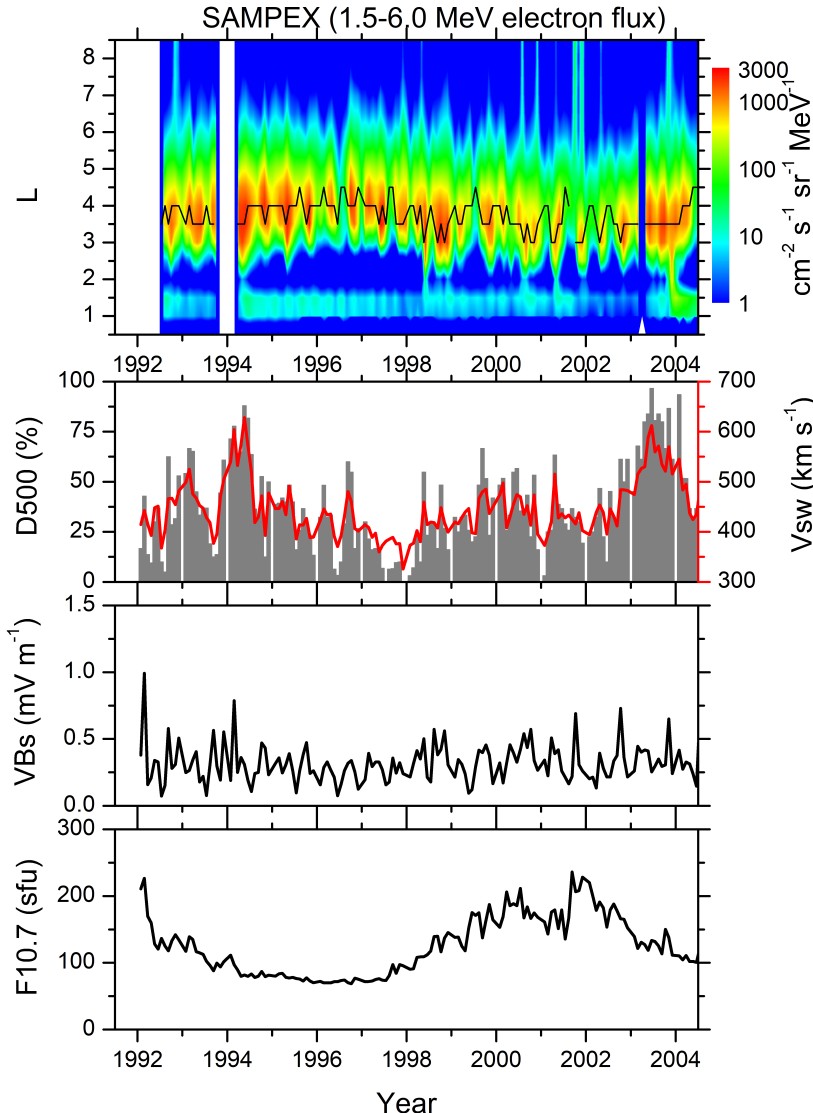

**Figure 1.** From top to bottom, the panels show the L-shell variation of monthly mean differential 1.5-6.0 MeV electron fluxes (legend on the right showing flux values corresponding to different colours) and L-shells corresponding to peak fluxes (black curve), monthly percentage of days with peak solar wind speed Vsw $\geq$ 500 $\mathrm{km\,s^{-1}}$ (legend on the left) and monthly mean Vsw (red curve, legend on the right), monthly mean VBs, and F10.7 solar flux for the years 1992 through 2004. F10.7 is expressed in solar flux unit (sfu), where a sfu is $10^{-22}\,\mathrm{W\,m^{-2}\,Hz^{-1}}$.

the belt at higher L-shells and flux decreases can be noted in the ascending and maximum phases of solar cycle 23. A close-

look in the fluxes reveals smaller-scale flux variations in each year around the heart of the outer belt. A similar smaller-scale variation is recorded in VBs.

Figure 2 shows Lomb-Scargle periodograms (Lomb, 1976; Scargle, 1982) of F10.7 solar flux, Vsw, D500, VBs and MeV electron fluxes at different L-shells, shown in Figure 1. As expected, F10.7 exhibits a single periodicity of ~11 years depicting solar activity cycle. Interestingly, Vsw has a broad peak around 9.5-year period with additional significant peaks at ~4.6 and ~3.2 year periods. The D500 exhibited a similar (to Vsw) periodogram. The coupling function VBs is independent of solar activity and has a significant period of ~0.5 years or ~6 months only.

The electron fluxes at different L-shells exhibit large variations in periodicity. At L = 1.0 (inner belt) electrons exhibit significant periods of ~8.9 and ~4.3 years. These seem to be associated with variations in Vsw (D500) with similar periods. However, it should be noted that electron measurement in the inner belt is largely contaminated by very energetic ($\geq$ 10-100 MeV) protons (see, e.g., Singer, 1958; Fennell et al., 2015; Selesnick et al., 2016, and referenes therein). For obvious reason, slot region electrons have no significant variation. At the inner edge of the outer belt (L = 3.0 and 3.5), electrons exhibit a significant periodicity of ~6 months, but no periodicity related to the ~11-year solar activity cycle. In the shells between L = 4.0 to L = 5.0, an ~11-year periodicity is accompanied by a prominent periodicity of ~6 months. The ~6-month periodicity in the electrons (for L = 3.0 to 5.0) can be attributed to the variations in the coupling function VBs. Electrons at L = 5.5 and 6.0 exhibit only a significant periodicity of ~11 years. At L > 6.0, there is no clear periodic variations in MeV electron fluxes.

Figure 2 clearly indicates varying solar activity and seasonal variations of the 1.5-6.0 MeV electrons at different L-shells, which can be attributed to different solar and magnetospheric drivers. This will be discussed later in the paper.

Figure 3 shows the year-month contour plots of monthly mean MeV electron fluxes at L = 3.5 and monthly mean VBs. The top panel shows the ratio of electron flux seasonal peaks in two halves of each year. This may give an estimate of the seasonal asymmetry of the electron fluxes. It should be noted that months of the peaks varied from year to year, which will be discussed below. Monthly mean F10.7 solar flux is repeated from Figure 1 for a reference of solar activity cycle. Similar analysis is performed for electrons at other L-shells, however they are not shown here to avoid repetition and save space.

While Figure 2 showed a ~6-month (semi-annual) component both in electron flux variation at L = 3.5 and in VBs variation, a clear year-to-year variation can be observed in Figure 3. In the year 1993, electron fluxes peak around May and August, while two peaks are observed during the months of May and November in 1994. In both cases, the first peaks are ~1.5-1.7 times higher than the second peaks. In April and October 1995, the two electron flux peaks are much more distinct and comparable in amplitude. The semi-annual variation is much weaker in 1996 with a peak in April ~0.5 times of that in October. In 1997, a large peak in fluxes can be noted in May with no prominent equinoctial peaks. In 1998, which is in the ascending phase of solar cycle 23, two distinct and comparable peaks are recorded during May and September. In 1999, while electron fluxes are much lower, two peaks can be noted in February and October. No clear seasonal feature can be inferred from 2000, while a solstice peak (May) is observed in 2001, and an October peak in 2002. A two-peak seasonal feature, with two distinct and comparable peaks in May and September, is again observed in 2003, in the descending phase of solar cycle 23. A consistent variation is observed in VBs with respect to month and year.

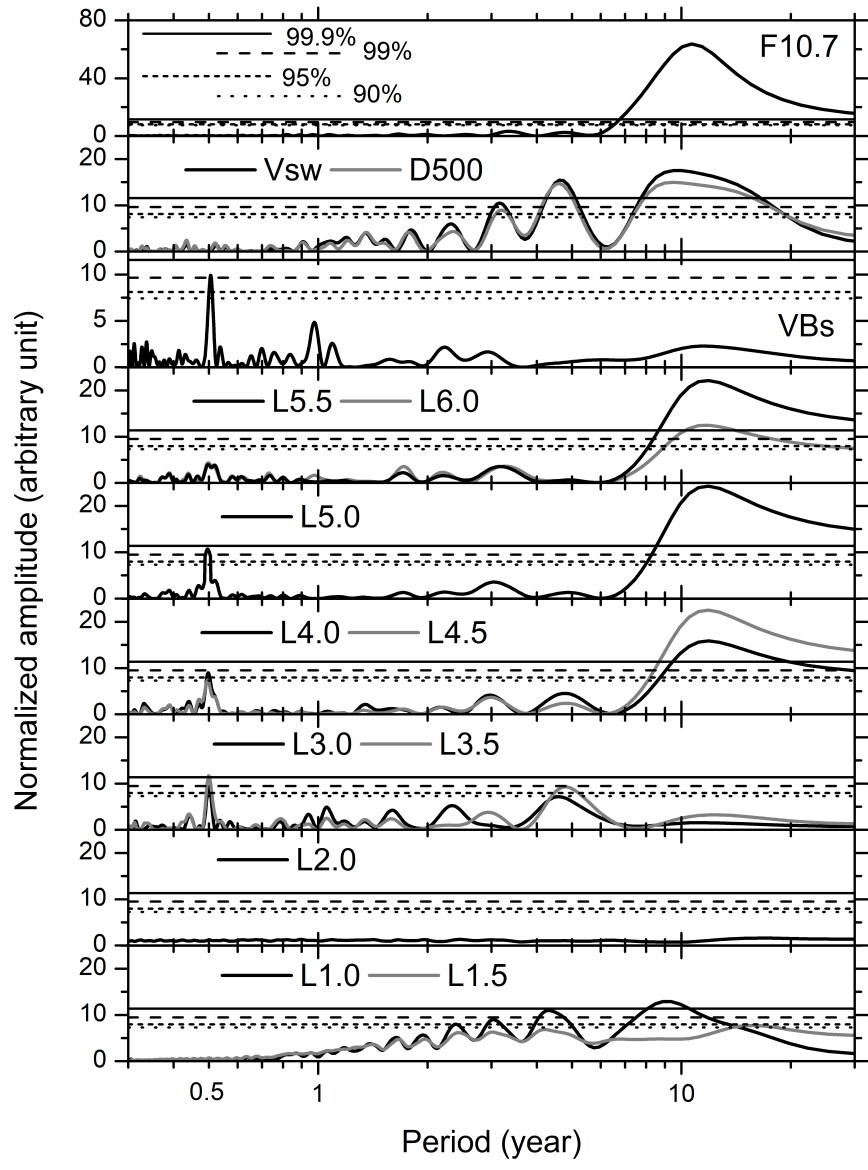

**Figure 2.** From top to bottom, the panels show Lomb-Scargle periodgrams of F10.7, Vsw and D500, VBs, and 1.5-6.0 MeV electron fluxes at different L-shells. The x-axis shows periods in year and the y-axis shows the normalized amplitudes in arbitrary unit. Confidence levels of the periodograms are shown in each panel by horizontal lines.

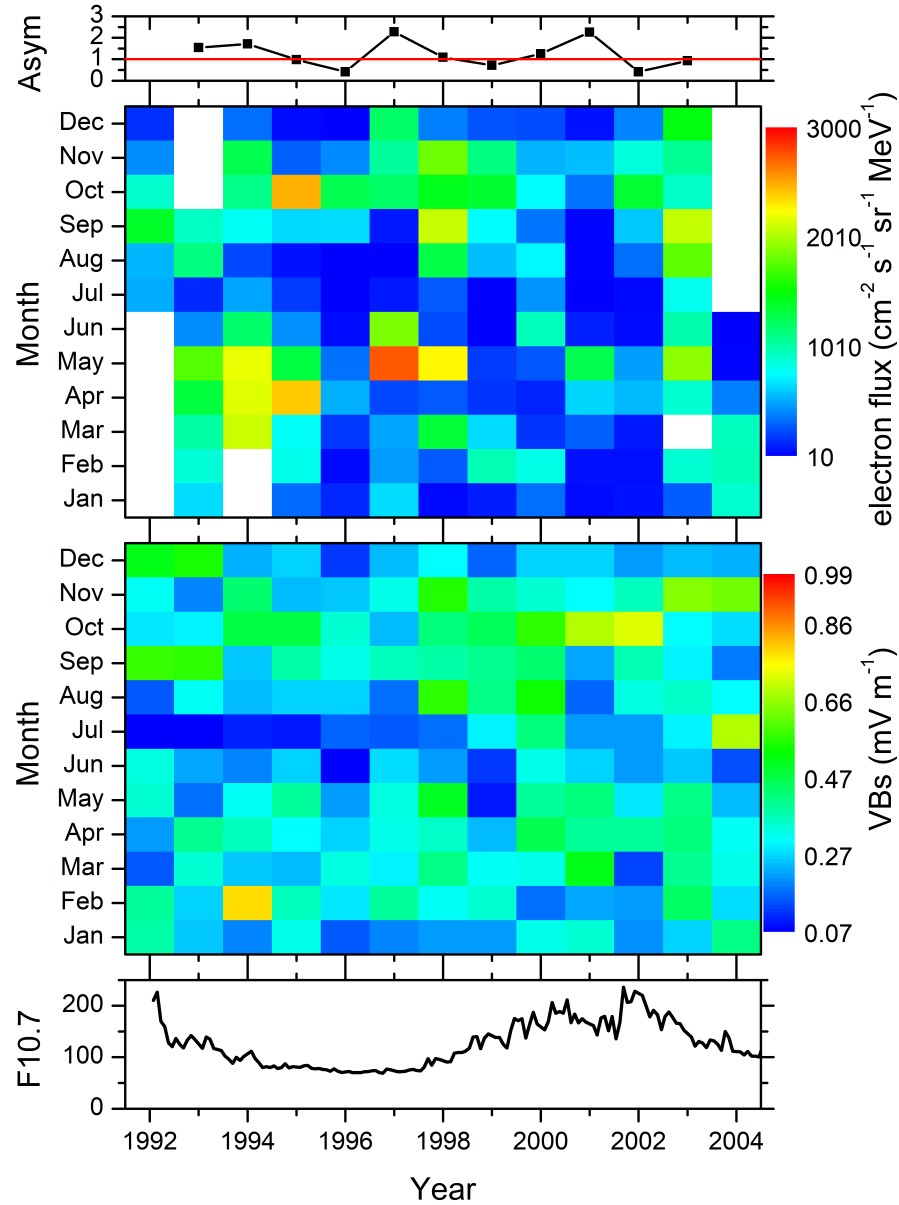

**Figure 3.** Contour plots in second and third panels show monthly mean 1.5-6.0 $MeV$ electron fluxes at L = 3.5 and monthly mean VBs, respectively during each month of the years 1992 through 2004. Top panel shows seasonal asymmetry (defined in text) and bottom panel shows the F10.7 solar flux during the same interval.

It can be concluded from the above analysis that while the 1.5-6.0 MeV electron fluxes at L = 3.5 exhibit mostly two peaks in a year, they are largely asymmetric in amplitude and they are not essentially equinoctial. The same conclusions were drawn for electrons at other L-shells.

## 3   Discussion and Conclusions

The results of this paper reveal that the MeV electrons in the Earth's outer zone radiation belt exhibit varying solar and seasonal features depending on the L-shells. No ∼11-year solar cycle trend was observed in the inner edge (L = 3.0-3.5) of the outer belt, where a dominating ∼6-month period was prominent. From L = 4.0 to 5.0, the ∼11-year solar cycle variation is accompanied by a secondary ∼6-month period. Electrons at L = 5.5 and 6.0 exhibit only a significant periodicity of ∼11 years, above which there is no clear periodic variations in MeV electron fluxes. These L-shell dependent seasonal and solar cycle features are reported for the first time. These are in contrast to previous studies (e.g., Baker et al., 1999; Li et al., 2001; Kanekal et al., 2010) that reported strong and "coherent" seasonal modulations of MeV electrons "throughout the entire outer zone" radiation belt.

Interestingly, no ∼6-month component was observed in solar wind speed Vsw, while it was most prominent in the solar wind-magnetospheric coupling function VBs which represents the interplanetary electric field under the condition of southward IMFs. This implies that the seasonal feature is due to magnetic configuration (Bs). In addition, the absence of a periodic component in Vsw below a period of a few years is of interest, and shows that the solar wind activity is intrinsically aperiodic on these time scales, so that the observed seasonal dependency can only be proper to the geospheric system (which is compatible with the usual explanation of the seasonal effect). The above result is consistent with previous results (e.g., Li et al., 2011) suggesting that HSS alone can not predict relativistic electron flux enhancements, but that fast solar wind and southward IMFs are the main requirements for electron enhancements. This makes VBs, involving both solar wind speed and southward IMF, an important factor controlling MeV electron variation.

However, present study involving multi-year analysis of seasonal features in MeV electrons and solar/magnetospheric drivers reveals that care should be taken in interpreting the ∼6-month periodicity obtained through periodogram analysis (present work) or superposed analysis of electrons in the radiation belt (previous reports). Yearly two peaks in the electron fluxes (between L = 3.0 and 5.0) are only sometimes observed around descending phase of the solar cycle. The peaks are largely asymmetric in nature. In addition, the peaks are essentially not equinoctial: sometimes the peaks are shifted to solstices and sometimes one annual peak is only observed. Clearly the ∼6-month periodicity in periodogram (and semi-annual variation) of the magnetospheric MeV electrons is an artifact arising from long-term data superposition in years. That the seasonal effects are of statistical in nature and they apply to the geoeffetiveness of the solar/interplanetary drivers are now well understood (see, Cliver et al., 2000, 2004; Nowada et al., 2009; Mursula et al., 2011; Cnossen and Richmond, 2012; Lockwood et al., 2020; Tsurutani et al., 2020, and references therein). However, for applying axial, equinoctial or geometrical hypothesis to discuss Earth's radiation belt importance of solar cycle period component and impact of Vsw should be considered. In addition,

125 radiation belt dynamics is strongly dependent on the energy of the electrons. Thus the energy dependence on the seasonal variations should also be investigated for a more complete understanding.

In summary, the L-shell dependent solar and seasonal features and so-called semi-annual variations of magnetospheric relativistic electrons require further attention. Dual satellite Van Allen Probes (Mauk et al., 2013) involving multi-energy observations of the radiation belts can be useful for further confirmation of the results obtained in the present work.

130 *Data availability.* Relativistic (MeV) electrons analyzed in this work is observed by the SAMPEX. These can be obtained from the Co-ordinated Data Analysis Web (CDAWeb) (https://cdaweb.gsfc.nasa.gov/cgi-bin/eval1.cgi). The solar wind and IMFs are obtained from the OMNI website (https://omniweb.gsfc.nasa.gov/)

*Author contributions.* RH developed the paper with original idea, data analysis and conclusion of the paper.

*Competing interests.* The author declares no competing interests.

135 *Acknowledgements.* The work is funded by the Science & Engineering Research Board (SERB), a statutory body of the Department of Science & Technology (DST), Government of India through Ramanujan Fellowship. I would like to thank Prof. Bruce T. Tsurutani for helpful scientific discussions.

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
