# Peer review of "Seasonal dependence of the Earth's radiation belt: new insights"

_Annales Geophysicae, 2020_

## Referee Comment (RC1) · Anonymous Referee #1 · 27 Oct 2020

This article presents an analysis of the periodicity of electron flux enhancements in the Earth radiation belts, and of its main solar wind drivers. Periodgrams are established, showing various periodicities (maily linked to the solar cycle and the seasonal periodicity), depending on the L-shells and for different solar wind parameters. Focusing on L=3.5, this articles then shows that the seasonal dependency can only be seen on multi-year statistics, and a large variability is shown from one year to another in the presence and position of flux peaks. While not surprising, these observations might not have been published earlier, and a carefull analysis of the year-wise variability of the electron outer belt is of interest to the community.

The language in this article is clear and concise, and the figures are clear, easily readable and appropriately described.

[Figure]

However, I have the following remarks concerning this article:

- Why was the L parameter used for this study? The L* parameter, which is an invariant of the motion of the particles, would certainly provide a clearer picture of the electron radiation belt dynamics, particularly at high L values.

- On line 20, the explained mechanism mostly applies to the outer radiation belt, and obviously not in or below the slot. This is confirmed by the provided periodgrams, but should be noted.

- On line 94, the fact that the VB parameter has a 6 month component that is not shared by Vsw is not surprising, since the seasonal periodicity is due to the magnetic configuration. The absence of of periodic component in Vsw below a period of a few years is of interest, and shows that the solar wind activity is intrinsically aperiodic on these time scales, so that the observed seasonal dependency can only be proper to the geospheric system (which is compatible with the usual explanation of the seasonal effect).

- On line 105, the article seems to imply that the current understanding of the seasonal effects (namely the equinoctial configuration of the magnetic field being linked to increased geoeffectiveness of the storms) does not explain the observations presented here, due to the variability of the observed peaks from one year to another. I think the community is aware that the seasonal effects are statistical in nature, since they act on the geoeffectiveness of the storms, and not on the occurences of the storms (which are aperiodic on short time scales, and have a solar-cycle period component, as shown in the plots of Vsw). The observed year-wise variability is expected with the classical model, which is not clear at all in this article. A more detailed and rigorous analysis of this variability would be of interest to the community, but the mere existence of this variability seems obvious.

---

## Author Response (AR1)

**Reply to the comments by Topical Editor**:

Dear Dr. Roussos,

I am submitting herewith the revised manuscript (MS No.: angeo-2020-62): "Seasonal dependence of the Earth's radiation belt: new insights". Thank you for valuable comments and suggestions. The manuscript is revised based on comments and suggestions by you and the Referee #1. The modifications are marked in the revised version with "track changes". I explain below how your suggestions are incorporated in the revision. I also include a response file listing all comments and suggestions by the referee and describing how they are incorporated in the revision.

Dear Dr. Hajra,

Thank you for submitting your manuscript "Seasonal dependence of the Earth's radiation belt: new insight". As you may have realised from the referee report, the reviewer considers that your study has some original aspects and its therefore worth considering for publication in our journal. However, I agree with the reviewer that discussion of your methods and results could be further enhanced. (Reply: Thank you. The manuscript is now revised based on all comments and suggestions by the reviewer.) On balance, I see very few references to results by the Van Allen Probes (VAP), which show much more dynamics in the slot region and also the inner boundary of the outer belt. The review paper by Baker et al. 2018 on space weather is included but in a generic way in the first paragraph of the manuscript (Reply: Thank you for the comment. We now add more references to the VAP study related to dynamics of the inner boundary of the outer belt.). Analysis of the inner belt is shown in Figure 2, but with little suspicion about its contamination by protons: VAP has shown that the spectrum there does not extend above 1.5 MeV (while the data shown in the manuscript are between 1.5 – 6 MeV) and part of the signal in older measurements may have been due to contamination (Reply: Thank you for the comment. Contamination of the inner belt by protons is now discussed in the manuscript.). Also, several studies of the outer belt show a strongly energy dependent evolution of the electron distributions in the 1.5 to 6 MeV range – how much can the energy dependence contribute to your variability measurements? (Reply: Thank you for raising this issue of energy dependence. In fact, further study using multi-energy observations by VAP can be useful for this. This is mentioned in the manuscript.)

Your effort to provide answers to the reviewer comments in the interactive discussion is nevertheless appreciated. To the extend these answers can be reviewed at an editorial level by myself, they seem sufficient for your manuscript to go into the next round of the peer review process. (Reply: Thank you)

I therefore encourage you to proceed with a submission of the revision along the lines of what you have already offered in the interactive discussion. You may use the same answers in addition to updating your manuscript accordingly for your formal revision submission, but of course you are free to make further changes if you consider this necessary. Please also consider my editorial comments before proceeding and include also a marked-up version of your revised manuscript where it is indicated which changes were made. (Reply: Thank you. The manuscript

is now revised based on comments and suggestions by you and the referee. The modifications in the manuscript are shown in the version with "track changes".)

I expect that your manuscript will be sent back to the reviewer for a follow-up review.

Kind regards,

Rajkumar Hajra

**Reply to the comments by Referee #1**:

I would like to thank the Referee #1 for carefully reading the manuscript and giving valuable comments and suggestions. The manuscript is now revised accordingly. I outline below how your comments and suggestions are incorporated in the revision.

This article presents an analysis of the periodicity of electron flux enhancements in the Earth's radiation belts, and of its main solar wind drivers. Periodgrams are established, showing various periodicities (mainly linked to the solar cycle and the seasonal periodicity), depending on the L-shells and for different solar wind parameters. Focusing on L=3.5, this articles then shows that the seasonal dependency can only be seen on multi-year statistics, and a large variability is shown from one year to another in the presence and position of flux peaks. While not surprising, these observations might not have been published earlier, and a careful analysis of the year-wise variability of the electron outer belt is of interest to the community.
- Reply: Thank you.

The language in this article is clear and concise, and the figures are clear, easily readable and appropriately described.
- Reply: Thank you.

However, I have the following remarks concerning this article:

- Why was the L parameter used for this study? The L* parameter, which is an invariant of the motion of the particles, would certainly provide a clearer picture of the electron radiation belt dynamics, particularly at high L values.
- Reply: I agree with you that the L* parameter (Roederer L parameter) would provide a clearer picture of electron radiation belt dynamics compared to the McIlwain L parameter for large L/L* > 5. However, for smaller L/L* the results will remain the same. I used the readily available L parameter, which was directly provided with the SAMPEX data. It should be noted that the L parameter has been widely used by SAMPEX scientists (references are provided in the manuscript). In addition, because most of the primary results presented in the work pertain to L < 5.0, it is felt that the L parameter is reasonable to use for this effort. I believe that this will not largely impact the results and interpretations.

- On line 20, the explained mechanism mostly applies to the outer radiation belt, and obviously not in or below the slot. This is confirmed by the provided periodograms, but should be noted.
- Reply: Thank you for the comment. That the described mechanism applies for outer zone radiation belt is now made clear in the manuscript.

- On line 94, the fact that the VB parameter has a 6-month component that is not shared by Vsw is not surprising, since the seasonal periodicity is due to the magnetic configuration. The absence of periodic component in Vsw below a period of a few years is of interest, and shows that the solar wind activity is intrinsically aperiodic on these time scales, so that the observed seasonal dependency can only be proper to the geospheric system (which is compatible with the usual explanation of the seasonal effect).
- Reply: Thank you for the comment. That the 6-month periodicity in VBs originated from magnetic configuration and that solar wind does not have any intrinsic seasonal variation are now discussed in greater detail in the revised manuscript.

- On line 105, the article seems to imply that the current understanding of the seasonal effects (namely the equinoctial configuration of the magnetic field being linked to increased geoeffectiveness of the storms) does not explain the observations presented here, due to the variability of the observed peaks from one year to another. I think, the community is aware that the seasonal effects are statistical in nature, since they act on the geoeffectiveness of the storms, and not on the occurrences of the storms (which are aperiodic on short time scales, and have a solar-cycle period component, as shown in the plots of Vsw). The observed year-wise variability is expected with the classical model, which is not clear at all in this article. A more detailed and rigorous analysis of this variability would be of interest to the community, but the mere existence of this variability seems obvious.
- Reply: Thank you for the comment. The discussion is now revised. The current understanding of the seasonal effects in terms of equinoctial configuration of the magnetic fields leading to increased geoeffectiveness are mainly based on studies of magnetic storms. And this is mostly statistical in nature. However, in order to discuss the L-shell distribution of radiation belt electrons, we need to consider the important role of the solar wind speed which is aperiodic on short time scales; the geomagnetic configuration cannot entirely explain the observations. This is because relativistic electrons are mainly associated with substorms and convection events during HILDCAAs, the latter exhibit strong associations with solar wind high-speed streams. In addition, HILDCAAs do not exhibit any semi-annual variation. These are now made clearer in the revised manuscript.

[revised manuscript text omitted]